# Environmental drivers of systematicity and generalisation in a situated agent

**Felix Hill**[1], **Andrew Lampinen**[3]*, **Rosalia Schneider**[1], **Stephen Clark**[1]
**Matthew Botvinick**[1,2], **James L. McClelland**[1,3] **& Adam Santoro**[1]

[1] DeepMind, London
[2] Gatsby Computational Neuroscience Unit, University College London
[3] Dept. of Psychology, Stanford University

## Abstract

The question of whether deep neural networks are good at generalising beyond their immediate training experience is of critical importance for learning-based approaches to AI. Here, we consider tests of out-of-sample generalisation that require an agent to respond to never-seen-before instructions by manipulating and positioning objects in a 3D Unity simulated room. We first describe a comparatively generic agent architecture that exhibits strong performance on these tests. We then identify three aspects of the training regime and environment that make a significant difference to its performance: (a) the number of object/word experiences in the training set; (b) the visual invariances afforded by the agent's perspective, or frame of reference; and (c) the variety of visual input inherent in the perceptual aspect of the agent's perception. Our findings indicate that the degree of generalisation that networks exhibit can depend critically on particulars of the environment in which a given task is instantiated. They further suggest that the propensity for neural networks to generalise in systematic ways may increase if, like human children, those networks have access to many frames of richly varying, multi-modal observations as they learn.

## 1 Introduction

Since the earliest days of research on neural networks, a recurring point of debate is whether neural networks exhibit generalisation beyond their training experience, in a *systematic* way (Smolensky, 1988; Fodor & Pylyshyn, 1988; Marcus, 1998; McClelland et al., 1987). This debate has been re-energized over the past few years, given a resurgence in neural network research overall (Lake & Baroni, 2017; Bahdanau et al., 2018; Lake, 2019). Generalisation in neural networks is *not* a binary question; since there are cases where networks generalise well and others where they do not, the pertinent research question is *when* and under *what conditions* neural networks are able to generalise. Here, we establish that a conventional neural-network-based agent exposed to raw visual input and symbolic (language-like) instructions readily learns to exhibit generalisation that approaches systematic behaviour, and we explore the conditions supporting its emergence. First, we show in a 3D simulated room that an agent trained to `find` all objects from a set and `lift` only some of them can `lift` withheld test objects never lifted during training. Second, we show that the same agent trained to `lift` all of the objects and `put` only some of them during training can `put` withheld test objects, zero-shot, in the correct location. That is, the model learns to re-compose known concepts (verbs and nouns) in novel combinations.

In order to better understand this generalisation, we conduct several experiments to isolate its contributing factors. We find three to be critical: (a) the number of words and objects experienced during training; (b) a bounded frame of reference or perspective; and (c) the diversity of perceptual input afforded by the temporal aspect of the agent's perspective. Crucially, these factors can be enhanced by situating an agent in a realistic environment, rather than an abstract, simplified setting. These results serve to explain differences between our findings and studies showing poor generalisation, where networks were typically trained in a supervised fashion on abstract or idealised stimuli from a

---

*Work carried out during internship at DeepMind.

single modality (e.g. Lake & Baroni, 2017). They also suggest that the human capacity to exploit the compositionality of the world, when learning to generalise in systematic ways, might be replicated in artificial neural networks if those networks are afforded access to a rich, interactive, multimodal stream of stimuli that better matches the experience of an embodied human learner (Clerkin et al., 2017; Kellman & Arterberry, 2000; James et al., 2014; Yurovsky et al., 2013; Anderson, 2003). Our results suggest that robust systematic generalisation may be an *emergent property* of an agent interacting with a rich, situated environment (c.f. McClelland et al., 2010).

## 1.1 SYSTEMATICITY AND GENERALISATION

Systematicity is the property of human cognition whereby "the ability to entertain a given thought implies the ability to entertain thoughts with semantically related contents" (Fodor & Pylyshyn, 1988). As an example of systematic thinking, Fodor & Pylyshyn (1988) point out that any human who can understand *John loves Mary* will also understand the phrase *Mary loves John*, whether or not they have heard the latter phrase before. Systematic generalisation (Plaut, 1999; Bahdanau et al., 2018; Lake et al., 2019) is a desirable characteristic for a computational model because it suggests that if the model can understand components (or words) in certain combinations, it should also understand the same components in different combinations. Note that systematic generalisation is also sometimes referred to as 'combinatorial generalisation' (O'Reilly, 2001; Battaglia et al., 2018). However, even human reasoning is often not perfectly systematic (O'Reilly et al., 2013). We therefore consider this issue from the perspective of generalisation, and ask to what degree a system generalises in accordance with the systematic structure implied by language.

Recent discussions around systematicity and neural networks have focused on the issue of how best to encourage this behaviour in trained models. Many recent contributions argue that generalising systematically requires inductive biases that are specifically designed to support some form of symbolic computation (such as graphs (Battaglia et al., 2018), modular-components defined by symbolic parses (Andreas et al., 2016; Bahdanau et al., 2018), explicit latent variables (Higgins et al., 2017) or other neuro-symbolic hybrid methods (Mao et al., 2019)). On the other hand, some recent work has reported instances of strong generalisation in the absence of such specific inductive biases (Chaplot et al., 2018; Yu et al., 2018; Lake, 2019). In the following sections, we first add to this latter literature by reporting several novel cases of emergent generalisation. Unlike in previous work, the examples that we present here involve tasks involving the manipulation of objects via motor-policies, as well as language and vision. This is followed by an in-depth empirical analysis of the environmental conditions that stimulate generalisation in these cases.

## 2  A MINIMAL MULTI-MODAL AGENT

All our experiments use the same agent architecture, a minimal set of components for visual (pixels) perception, language (strings) perception, and policy prediction contingent on current and past observations. The simplicity of the architecture is intended to emphasize the generality of the findings; for details see App C.

**Visual processing** Visual observations at each timestep come in the form of a $W \times H \times 3$ real-valued tensor ($W$ and $H$ depend on the particular environment), which is processed by a 3-layer convolutional network with $64, 64, 32$ channels in the first, second and third layers respectively. The flattened output of this network is concatenated with an embedding of the language observation.

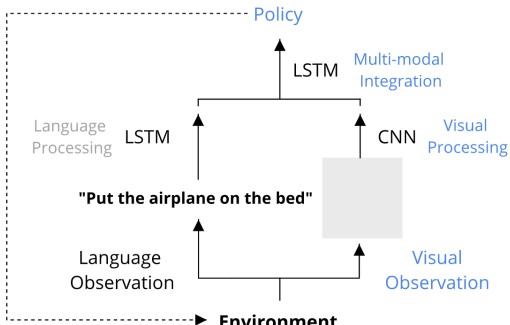

**Figure 1:** Schematic of the architecture used in all experiments. The blue components show some critical differences that differentiate it from more abstract studies that reported failures in systematic generalization.

**Language processing** Language instructions are received at every timestep as a string. The agent splits these on whitespace and processes them with a (word-level) LSTM network with hidden state

size 128. The final hidden state is concatenated with the output of the visual processor to yield a multimodal representation of the stimulus at each timestep.

**Memory, action and value prediction** The multimodal representation is passed to a 128-unit LSTM. At each timestep, the state of this LSTM is multiplied by a weight matrix containing $A \times 128$ weights; the output of this operation is passed through a softmax to yield a distribution over actions ($A$ is the environment-dependent size of the action set). The memory state is also multiplied by a $1 \times 128$ weight matrix to yield a value prediction.

**Training algorithm** We train the agent using an importance-weighted actor-critic algorithm with a central learner and distributed actors (Espeholt et al., 2018).

## 3 DEMONSTRATING GENERALISATION

A key aspect of language understanding is the ability to flexibly combine predicates with arguments; verb-noun binding is perhaps the canonical example. Verbs typically refer to processes and actions, so we study this phenomenon in a 3D room built in the Unity game engine.[1] In this environment, the agent observes the world from a first-person perspective, the Unity objects are 3D renderings of everyday objects, the environment has simulated physics enabling objects to be picked-up, moved and stacked, and the agent's action-space consists of 26 actions that allow the agent to move its location, its field of vision, and to grip, lift, lower and manipulate objects. Executing a simple instruction like `find a toothbrush` (which can be accomplished on average in six actions by a well-trained agent in our corresponding grid world) requires an average of around 20 action decisions.

### 3.1 A GENERAL NOTION OF LIFTING

*Lifting* is a simple motor process that corresponds to a verb and is easily studied in this environment. We consider the example instruction `lift a helicopter` to be successfully executed if the agent picks up and raises the helicopter above a height of 0.5m for 2 seconds, a sequence which requires multiple actions once the helicopter is located. Similarly, the instruction `find a helicopter` is realised if the agent moves within two metres of a helicopter and fixes its gaze (as determined by a ray cast from the agent's visual field) while remaining within the two metre proximity for five timesteps *without lifting the object during this time*.

To measure how general the acquired notion of lifting is, we trained the agent to `find` each of a set $X = X_1 \cup X_2$ of different objects (allowing the agent to learn to ground objects to their names) and to `lift` a subset $X_1$ of those objects, in trials in a small room containing two objects positioned at random (one 'correct' according to the instruction, and one 'incorrect'). The agent receives a positive reward if it finds or lifts the correct object, and the episode ends with no reward if the agent finds or lifts the incorrect object. We then evaluate its ability to extend its notion of lifting (zero-shot) to objects $x \in X_2$. In a variant of this experiment, we trained the agent to lift all of the objects in $X$ when referring to them by their color (`lift a green object`), and to find all of the objects in $X$ according to either shape or color (`find a pencil` or `find a blue object`). We again tested on whether the agent lifted objects $x \in X_2$ according to their shape (so, the test trials in both variants are the same). As shown in Fig 2(a), in both variants the agent generalises with near-perfect accuracy. The agent therefore learned a notion of what it is to lift an object (and how this binds to the word *lift*) with sufficient generality that it can, without further training, apply it to novel objects, or to familiar objects with novel modes of linguistic reference.

### 3.2 A GENERAL NOTION OF PUTTING

We took our study of predicate-object binding further by randomly placing two large, flat objects (a bed and a tray) in the room and training an agent to place one of three smaller objects on top. As before, the agent received a positive reward if it placed the correct small object on the bed or the tray according to the instruction. If it placed an incorrect object on the bed or tray, the episode ended with no reward. To test generalisation in this case, we trained the agent to `lift` each of a set $X = X_1 \cup X_2$ of smaller objects (as in the previous experiment) and then to put some

---

[1]http://unity3d.com

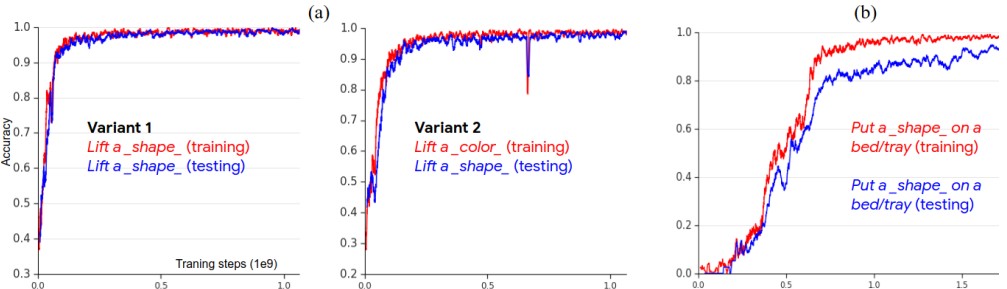

**Figure 2:** Test and training performance as an agent learns zero-shot predicate-argument binding for (a) 'lifting' in two variants: (1) training instructions are e.g. `find a spaceship`, `find/lift a pencil`, testing instructions are e.g. `lift a spaceship`. (2) training instructions are e.g. `find a spaceship/lift a green object`, testing instructions are e.g. `lift a spaceship`. (b) 'putting': training instructions are e.g. `put a spaceship on the bed`, `lift a pencil`, testing instructions are e.g. `put a pencil on the bed`.

subset $X_1 \subset X$ of those objects on both the bed and the tray as instructed. We then measured its performance (with no further learning) in trials where it was instructed to put objects from $X_2$ onto either the bed or the tray. Surprisingly, we found that the agent was able to place objects with over 90% accuracy onto the bed or the tray zero-shot as instructed (Fig 2(b)). This generalisation requires the agent to bind its knowledge of objects (referred to by nouns, and acquired in the lifting trials) to a complex control process (acquired in the training putting trials) – involving locating the correct target receptacle, moving the object above it (avoiding obstacles like the bed-head) and dropping it gently. An important caveat here is that control in this environment, while finer-grained than in many common synthetic environments, is far simpler than the real world; in particular the process required to lift an object does not depend on the shape of that object (only its extent, to a degree). Once the objects are held by the agent, however, their shape becomes somewhat more important and placing something on top of something else, avoiding possible obstacles that could knock the object from the agent's grasp, is a somewhat object-dependent process.

## 4 UNDERSTANDING THE DRIVERS OF GENERALISATION

### 4.1 NUMBER OF TRAINING INSTRUCTIONS

To emphasize most explicitly the role of the diversity of training experience in the emergence of systematic generalisation, we consider the abstract notion of *negation*. Rumelhart et al. (1986) showed that a two-layer feedforward network with sufficient hidden units can learn to predict the negation of binary patterns. Here, we adapt this experiment to an embodied agent with continuous visual input and, importantly, focus not only learning, but also generalisation. We choose to consider negation since it is an example of an operator for which we found that, for our standard environment configuration, our agent unequivocally fails to exhibit an ability to generalize in a systematic way.

To see this, we generated trials in the Unity room with two different objects positioned at random and instructions of the form `find a box` (in which case the agent was rewarded for locating and fixing its gaze on the box) and `find` [something that is] `not a box` (in which case there was a box in the room but the agent was rewarded for locating the other object). Like Rumelhart et al. (1986), we found that it was unproblematic to train an agent to respond correctly to both positive and negative *training* inputs. To explore generalisation, we then trained the agent to follow instructions `find a` $x$ for all $x \in X$ and negated instructions `find a not` $x$ for only $x \in X_1$ (where $X = X_1 \cup X_2$, and $X_1 \cap X_2 = \emptyset$), and tested how it interpreted negative commands for $x \in X_2$.

When $X_1$ contained only a few objects ($|X_1| = 6$), the agent interpreted negated instructions (involving objects from $X_2$) with *below chance* accuracy. In other words, for objects $x_2 \in X_2$, in response to the instruction `find a not` $x_2$, the agent was more likely to find the object $x_2$ than the correct referent of the instruction. This is an entirely un-systematic interpretation of the negation

predicate.[2] Interestingly, however, for ($|X_1| = 40$) the agent achieved above chance interpretation of held-out negated instructions, and for ($|X_1| = 100$) performance on held-out negated instructions increased to 0.78 (Table 1).

| Training set | Train accuracy | Test accuracy |
|---|---|---|
| 6 words | 1.00 | 0.40 |
| 40 words | 0.97 | 0.60 |
| 100 words | 0.91 | 0.78 |

**Table 1:** Accuracy extending a negation predicate to novel arguments (test accuracy) when agents are trained to negate different numbers of words/objects.

These results show that, for a test set of fixed size, the degree of systematicity exhibited by an agent when generalizing can grow with the variety of words/objects experienced in the training set, even for highly non-compositional operators such as negation. Of course, the mere fact that larger training sets yield better generalisation in neural networks is not novel or unexpected. On the other hand, we find the emergence of a logical operator like negation in the agent in a reasonably systematic way (noting that adult humans are far from perfectly systematic (Lake et al., 2019))), given experience of 100 objects (again, not orders of magnitude different from typical human experience), to be notable, particularly given the history of research into learning logical operators in connectionist models and the importance of negation in language processing (Steedman, 1999; 2011). In what follows, we complement these observations by establishing that not only the amount, but also the type of training data (or agent experience) can have a significant impact on emergent generalisation.

## 4.2 3D VERSUS 2D ENVIRONMENT

(a)            (b)

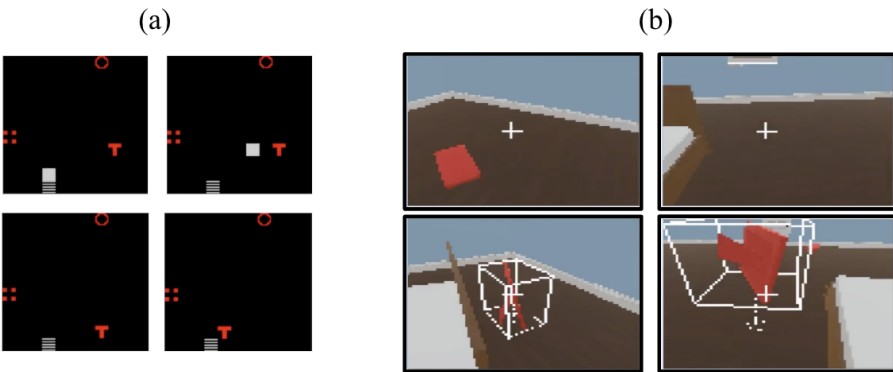

**Figure 3:** Screenshots from the agent's perspective of an episode in the grid-world and Unity environments. In both cases the instruction is `put the picture frame on the bed`.

Our first observation is that the specifics of the environment (irrespective of the logic of the task) can play an important role in emergent generalisation. To show this most explicitly we consider a simple *color-shape* generalisation task. Prior studies in both 2D and 3D environments have shown that neural-network-based agents can correctly interpret instructions referring to both the color and shape of objects (*find a red ball*) zero-shot when they have never encountered that particular combination during training (Chaplot et al., 2018; Hill et al., 2017; Yu et al., 2018). We replicate the demonstration of Hill et al. (2017) in the 3D DeepMind-Lab environment (Beattie et al., 2016), and for comparison implement an analogous task in a 2D grid-world (compare Fig 3 (a) and Fig 4).

As in the original experiments, we split the available colors and shapes in the environment into sets $S = \mathbf{s} \cup \hat{\mathbf{s}}$ and $C = \mathbf{c} \cup \hat{\mathbf{c}}$. We then train the agent on episodes with instructions sampled from one of the sets $\mathbf{c} \times \mathbf{s}$, $\hat{\mathbf{c}} \times \mathbf{s}$ or $\mathbf{c} \times \hat{\mathbf{s}}$, and, once trained, we evaluate its performance on instructions from $\hat{\mathbf{c}} \times \hat{\mathbf{s}}$. All episodes involve the agent in a small room faced with two objects, one of which matches the description in the instruction. Importantly, both the color and the shape word in the instruction are needed to resolve the task, and during both training and testing the agent faces trials in which

---

[2]We suspect in such cases that the agent is simply learning a non-compositional interpretation during training along the lines of `not` $x$ referring to the set $\{y \in X_1 : x \neq y\}$.

the confounding object either matches the color of the target object or the shape of the target object. While it was not possible to have exactly the same shapes in set $S$ in both the grid world and the 3D environment, the size of $C$ and $S$ and all other aspects of the environment engine were the same in both conditions. As shown in Table 2 (top), we found that training performance was marginally worse on the 3D environment, but that test performance in 3D ($M = 0.97, SD = 0.04$) was six percentage points higher than in 2D ($M = 0.91, SD = 0.08$) (a suggestive but not significant difference given the small sample of agents; $t(8) = 1.38, p = 0.20$).

To probe this effect further, we devised an analogue of the 'putting' task (Section 3.2) in the 2D grid-world. In both the 3D and 2D environments, the agent was trained on 'lifting' trials, in which it had to visit a specific object, and on 'putting' trials, in which it had to pick up a specific object and move it to the bed. To match the constraints of the grid-world, we reduced the total global object set in the Unity room to ten, allocating three to both lifting and putting trials during training, and the remaining seven only to lifting trials. The evaluation trials then involved putting the remaining seven objects on the bed (see Figure 3 for an illustration of the two environments). While the grid-world and the 3D room tasks are identical at a conceptual (and linguistic) level, the experience of an agent is quite different in the two environments. In the grid world, the agent observes the entire environment (including itself) from above at every timestep, and can move to the $81 = 9 \times 9$ possible locations by choosing to move up, down, left or right. To lift an object in the grid-world, the agent simply has to move to the square occupied by that object, while 'putting' requires the agent to lift the object and then walk to the square occupied by the white (striped) bed.

As shown in Table 2 (bottom), agents in both conditions achieved near perfect performance in training. However, on test trials, performance of the agent in the Unity room ($M = 0.63, SD = 0.06$) was significantly better than the agent in the grid world ($M = 0.40, SD = 0.14$); $t(8) = 3.48, p < 0.005$. In failure cases, the agent in the grid world can be observed exhibiting less certainty, putting the wrong object on the bed or running out of time without acting decisively with the objects.

|  | Train accuracy | Test accuracy |
|---|---|---|
| **Color-shape task** | | |
| Grid world | 0.99 | 0.91 ±0.09 |
| 3D room (DM-Lab) | 0.98 | 0.97 ±0.04 |
| | | |
| **Putting task** | | |
| Grid world | 0.99 | 0.40 ±0.14 |
| Grid world, scrolling | 0.93 | 0.60 ±0.14 |
| 3D room (Unity) | 0.99 | 0.63 ±0.06 |

**Table 2:** Tests of systematic generalisation in 2D and 3D environments; five randomly-initialized agent replicas in each condition.

### 4.3 VISUAL INVARIANCES IN AGENTS' PERSPECTIVES

To understand more precisely why agents trained in 3D worlds generalise better, we ran a further condition in which we gave the agent in the grid-world an ego-centric perspective, as it has in the 3D world. Specifically, we adapted the grid-world so that the agent's field of view was $5 \times 5$ and centred on the agent (rather than $9 \times 9$ and fixed). While this egocentric constraint made it harder for the agent to learn the training task, performance on the test set improved significantly $t(8) = 2.35, p < 0.05$, accounting for most of the difference in generalisation performance between the 3D world and the (original) 2D world. This suggests that the visual invariances introduced when bounding the agent's perspective, which in our case was implemented using an ego-centric perspective, may improve an agent's ability to factorise experience and behaviour into chunks that can be re-used effectively in novel situations. The relevant difference is likely that the agent is always in an invariant location in the visual input, which reduces some difficulty of perception, rather than the fact that the agent is centred per se. See Appendix D for control experiments showing that partial-observability alone does not induce the same boost to generalisation, nor does a difference in the information available from a single frame.

### 4.4 TEMPORAL ASPECT OF PERCEPTION

Egocentric vs. allocentric vision is not the only difference between a grid-world and a 3D world. Another difference is that, in the 3D world, the agent experiences a much richer variety of (highly

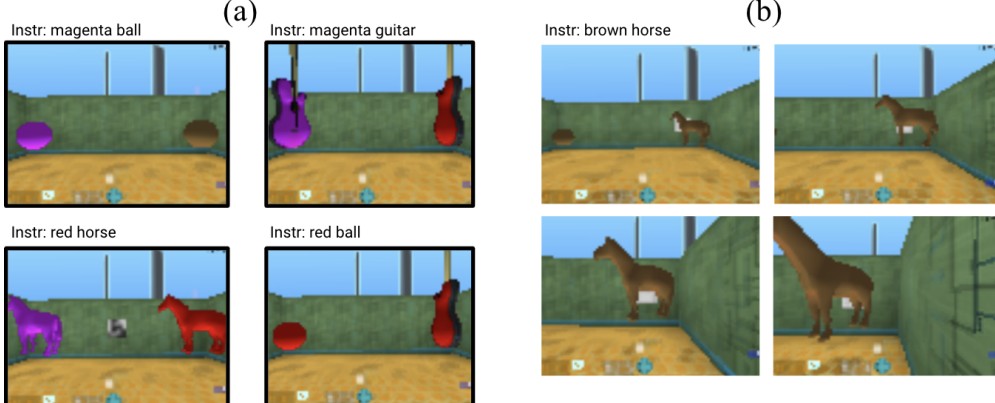

**Figure 4:** (a) Four (independent) training inputs from the training data for the classifier model. (b) Four timesteps of input from a single episode for the situated agent.

correlated) visual stimuli in any particular episode. To isolate the effect of this factor on generalisation, we return to the color-shape task, which is appropriate for the present experiment because a single train/test experiment (defined in terms of the instructions and objects that the agent experiences) can be constructed either as an interactive MDP for a situated agent or as a supervised classification problem.

In the **vision+language classifier** condition, a supervised model must predict either *left* or *right* in response to a still image (the first frame of an episode) of two objects and a language instruction of the form *find a red ball*. In the **agent** condition, our situated RL agent begins an episode facing two objects and is trained to move towards and bump into

| Regime | Train accuracy | Test accuracy |
|---|---|---|
| Classifier | 0.95 | 0.80 ±0.05 |
| Agent | 1.00 | 1.00 ±0.00 |

**Table 3:** generalisation accuracy achieved by a vision-and-language classifier trained on single screenshots versus a situated agent trained in the DMLab environment.

the object specified by the instruction. Importantly, the architecture for the vision+language classifer and the agent were identical except that the final (action and value-prediction) layer in the agent is replaced by a single linear layer and softmax over two possible outcomes in the classifier.

On the same set of training instructions, we trained the classifier to maximise the likelihood of its object predictions, and the agent to maximise the return from selecting the correct object. As shown in Table 3, the accuracy of the classifier on the training instructions converged at $0.95$ compared to $1.0$ for the agent (which may be explained by greater difficulty in detecting differences between objects given a more distant viewpoint). More importantly, performance of the classifier on test episodes ($M = 0.80$, $SD = 0.05$) was significantly worse than that of the agent ($M = 1.00$, $SD = 0.00$); $t(8) = 8.61$, $p < 0.0001$. We conclude that an agent that can move its location and its gaze (receiving richer variety views for a given set of object stimuli) learns not only to recognize those objects better, but also to generalise understanding of their shape and color with greater systematicity than an agent that can only observe a single snapshot for each instruction.[3] This richer experience can be thought of as effectively an implicit form of data augmentation.

### 4.5 THE ROLE OF LANGUAGE

Thus far, we have considered tasks that involve both (synthetic) language stimuli and visual observations, which makes it possible to pose diverse and interesting challenges requiring systematic behaviour. However, this approach raises the question of whether the highly regular language experienced by our agents during training contributes to the consistent systematic generalisation that we observe at test time. Language can provide a form of supervision for how to break down the world

---

[3]See Yurovsky et al. (2013) for a discussion of the importance of this factor in children's visual and word-learning development.

and/or learned behaviours into meaningful sub-parts, which in turn might stimulate systematicity and generalisation. Indeed, prior work has found that language can serve to promote compositional behaviour in deep RL agents (Andreas et al., 2017).

To explore this hypothesis, we devised a simple task in the grid world that can be solved either with or without relying on language. The agent begins each episode in a random position in the grid containing eight randomly-positioned objects, four of one color and shape and four of another. In each episode, one of the object types was randomly designated as 'correct' and the agent received a positive +1 reward for collecting objects of this type. The episode ended if the agent collected all of the correct objects (return = +4) or if it collected a single incorrect object (reward = 0). Without access to language, the optimal policy is to simply select an object type at random and then (if possible) the remainder of that type (which returns 2 on average). In the language condition, however, the target object type (e.g. 'red square') was named explicitly, so the agent could learn to achieve a return of 4 on all episodes. To test generalisation in both conditions, as in the color-shape reference expressions described above, we trained the agent on episodes involving half of the possible color-shape combinations (as correct or distractor objects) and tested it on episodes involving the other half. Note that, both during training and testing, in all episodes the incorrect object differed from the correct object in terms of either color or shape, but not both (so that awareness of both color and shape was necessary to succeed in general).

For fair comparison between conditions, when measuring agent performance we consider performance only on episodes in which the first object selected by the agent was correct. As shown in Figure 5, the non-linguistic agent performed worse on the training set (when excluding the 50% of trails in which it failed on the first object). Performance on the test episodes was also worse overall for the non-linguistic agent, but by a similar amount to the discrepancy on the training set (thus the *test error* was similar in both conditions). With increasing training, we observed this discrepancy to diminish to a negligible amount (Figure 5, right). Importantly, both linguistic and non-linguistic agents exhibited test generalisation that was substantially above chance. While not conclusive, this analysis raises the possibility that language may not be playing a significant role (and is certainly not the unique cause) of the systematic generalisation that we have observed emerging in other experiments.

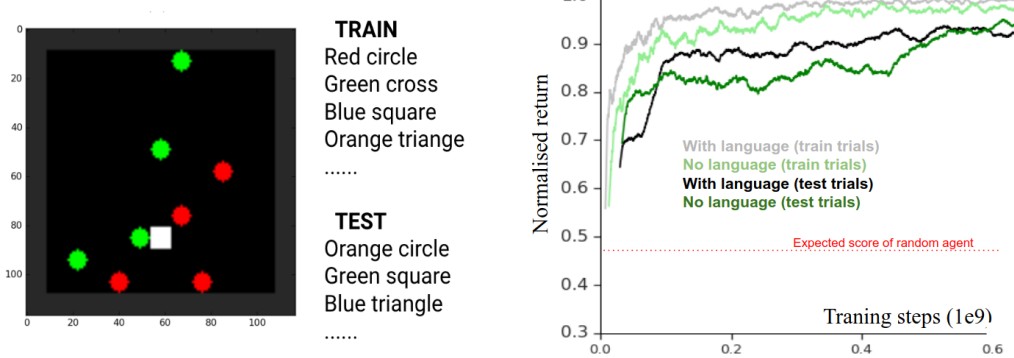

**Figure 5: Left** An episode of the grid-world task that can be posed with or without language. **Right** Normalised episode returns on training and evaluation trials as the agent learns, adjusted to ignore episodes where the first object that the agent collects is incorrect.

## 5 DISCUSSION

We have shown that a neural-network-based agent with standard architectural components can learn to execute goal-directed motor behaviours in response to novel instructions zero-shot. This suggests that, during training, the agent learns not only how to follow training instructions, but also general information about how word-like symbols compose and how the combination of those words affects what the agent should do in its world. With respect to generalisation, our findings contrast with other recent contributions suggesting that neural networks do not generalise well in these settings (Lake, 2019; Bahdanau et al., 2018), and are more aligned with older empirical analyses of neural networks (McClelland et al., 1987; Frank, 2006). Our work builds on those earlier studies

by considering not only the patterns or functions that neural networks can learn, but also how they compose familiar patterns to interpret entirely novel stimuli. Relative to more recent experiments on color-shape generalisation (Higgins et al., 2017; Chaplot et al., 2018; Yu et al., 2018), we study a wider range of phenomena, involving abstract modifying predicates (negation) and verbs ('to lift', 'to put') corresponding to complex behaviours requiring approximately 50 movement or manipulation actions and awareness of objects and their relations. By careful experimentation, we further establish that visual invariances afforded by a bounded frame of reference, in our case implemented using a first-person perspective of an agent acting over time, plays an important role in the emergence of this generalisation.

An obvious limitation of our work, which technology precludes us from resolving at present, is that we demonstrate the impact of realism on generalisation by comparing synthetic environments of differing degrees of realism, while even our most realistic environment is considerably more simple and regular than the world itself. This is in keeping with a long history of using simulation for the scientific analysis of learning machines (Minsky & Papert, 1969; Marcus, 1998). Nonetheless, as the possibilities for learning in situated robots develop, future work should explore whether the generalisation that we observe here is even closer to perfectly systematic under even more realistic learning conditions. An additional limitation of our work is that – much like closely-related work (Lake & Baroni, 2017; Bahdanau et al., 2018) – we do not provide a nomological explanation for the generalisation (or lack thereof) that we observe. Depending on the desired level of analysis, such an explanation may ultimately come from theoretical work on neural network generalisation (Arora et al., 2018; Lampinen & Ganguli, 2018; Allen-Zhu et al., 2018; Arora et al., 2019). Our focus here, however, is on providing careful experiments that isolate the significance of environmental factors on systematicity and to provide measures of the robustness (i.e. variance) of these effects.

We also emphasize that our agent in no way exhibits complete systematicity. The forms of generalisation it exhibits do not encompass the full range of systematicity of thought/behaviour that one might expect of a mature adult human. However, this is unsurprising given that our network has substantially less experience than an adult human, and learns in a substantially simpler environment. It is also worth noting that none of our experiments reflect the human ability to learn (rather than extend existing knowledge) quickly (as in, e.g. Lake et al. (2015); Botvinick et al. (2019)). Finally, depending on one's perspective, it is always possible to characterise the generalisation we describe here as interpolation. Rather than engage with this thorny philosophical distinction (which touches upon, e.g., the problem of induction (Vickers, 2009)), we simply emphasize that in our experiments there is a categorical difference between the data on which the agent was trained and test stimuli to which they respond. Finally, our experiments concur with a message expressed most clearly by Anderson's famous contribution in *Science* (Anderson, 1972). Conclusions that are reached when experimenting with pared-down or idealised stimuli may be different from those reached when considering more complex or naturalistic data, since the simplicity of the stimuli can stifle potentially important emergent phenomena. We suggest that strong generalisation may emerge when an agent is situated in a rich environment, rather than a simple, abstract setting.

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

## A  LIMITATIONS

Like a much recent AI research, we experiment with an agent learning and behaving in a simulated world, with the obvious limitation that *the data that the agent experiences are not 'real'*. Our paradigm does not directly reflect challenges in robotics, natural language processing, or indeed human learning and cognition. On the other hand, we note the long history of extrapolating from results on models in simulation to draw conclusions about AI in general (Minsky & Papert, 1969). The simulated experience of our agents in this study is in many ways more realistic than the idealised, symbolic data employed in those influential studies. Moreover, our results suggest that systematic generalization emerges most readily in the more realistic of the (simulated) environments that we consider, which is at least suggestive that the class of model that we consider here might behave in similar ways when moving towards learning in the real world.

Another clear limitation of our work is that – much like recent work that emphasises the failures of neural networks – we do not provide a nomological explanation for the generalization (or lack thereof) that we observe. That is, we do not provide give a precise answer to the question of *why?* systematic generalisation necessarily emerges. Depending on the desired level of explanation, such a question may ultimately be resolved by promising theoretical analyses of neural net generalization (Arora et al., 2018; Lampinen & Ganguli, 2018; Allen-Zhu et al., 2018; Arora et al., 2019), and some more general intuitions can be gleaned from developmental psychology (Clerkin et al., 2017; Kellman & Arterberry, 2000; James et al., 2014; Yurovsky et al., 2013). Our focus here, however, is on providing careful experiments that isolate the significance of environmental factors on systematicity and to provide measures of the robustness (i.e. variance) of these effects.

## B    EXPERIMENT DETAILS

### B.1    ENVIRONMENT DETAILS

At each timestep the agent can take one of 26 actions that allow it to move its body or head (direction of gaze), to grab objects that are in range, and to move or rotate held objects in various directions. An object is held while the agent executes actions with the GRAB prefix, and is dropped once the agent emits an action without the GRAB prefix. A visual aid (a white box) appears in the agent's view around any object that is within range of being grabbed, and when an object is held and lifted, a vertical line is projected below the object to assist in placing it (the limits of depth perception with monocular vision make placing objects very challenging otherwise).

| Body movement actions | Movement and grip actions | Object manipulation |
|---|---|---|
| NOOP | GRAB | GRAB + SPIN_OBJECT_RIGHT |
| MOVE_FORWARD | GRAB + MOVE_FORWARD | GRAB + SPIN_OBJECT_LEFT |
| MOVE_BACKWARD | GRAB + MOVE_BACKWARD | GRAB + SPIN_OBJECT_UP |
| MOVE_RIGHT | GRAB + MOVE_RIGHT | GRAB + SPIN_OBJECT_DOWN |
| MOVE_LEFT | GRAB + MOVE_LEFT | GRAB + SPIN_OBJECT_FORWARD |
| LOOK_RIGHT | GRAB + LOOK_RIGHT | GRAB + SPIN_OBJECT_BACKWARD |
| LOOK_LEFT | GRAB + LOOK_LEFT | GRAB + PUSH_OBJECT_AWAY |
| LOOK_UP | GRAB + LOOK_UP | GRAB + PULL_OBJECT_CLOSE |
| LOOK_DOWN | GRAB + LOOK_DOWN | |

**Table 4:** Action space of the agent in the 3D room.

### B.2    LIFTING, PUTTING AND NEGATION EXPERIMENTS

For lifting, putting and negation experiments, we divided the shapes into two sets, $X_1$ and $X_2$, to test generalisation to unseen combinations. These sets vary according to the experiment - see Tables 5 and 6.

In the negation experiment, we tested three sets of different sizes for $X_1$, and keeping $X_2$ always the same size. Table 7 shows the set division in each experiment.

### B.3    2D X 3D EXPERIMENTS

As explained in Sections 4.2 and 4.4, we considered a color-shape generalization task in DeepMind Lab (Beattie et al., 2016). Table 8 details the separation of objects in the sets $S = \mathbf{s} \cup \hat{\mathbf{s}}$ and $C = \mathbf{c} \cup \hat{\mathbf{c}}$.

### B.4    GRID-WORLD FINDING & PUTTING COMPARISON, AND LANGUAGE LESION

The grid-world environment consisted of a $9 \times 9$ room surrounded by a wall of width 1 square, for a total size of $11 \times 11$. These squares were rendered at a $9 \times 9$ pixel resolution, for a total visual input size of $99 \times 99$. The agent had 4 actions available, corresponding to moving one square in each of the 4 cardinal directions. The agent was given a time limit of 40 steps for each episode, which sufficed

| Sets | |
|---|---|
| $X_1$ | basketball, book, bottle, candle, comb, cube block, cuboid block, cushion fork, football, glass, grinder, hairdryer, headphones, knife, mug, napkin, pen, pencil, picture frame, pillar block, plate, potted plant, roof block, rubber duck, scissors, soap, soap dispenser, sponge, spoon, table lamp |
| $X_2$ | boat, bus, car, carriage, helicopter, keyboard, plane, robot, rocket, train, racket, vase |
| Color | aquamarine, blue, green, magenta, orange, purple, pink, red, white, yellow |

**Table 5:** Object/Color sets in the Lifting experiment.

| Sets | |
|---|---|
| $X_1$ | soap, picture frame, comb, mug, headphones, candle, cushion, potted plant, glass, tape dispenser, basketball, mirror, book, bottle, candle, grinder, rubber duck, soap dispenser, glass, pencil |
| $X_2$ | toy robot, teddy, hairdryer, plate, sponge, table lamp, toilet roll, vase, napkin, keyboard |
| Color | aquamarine, blue, green, magenta, orange, purple, pink, red, white, yellow |

**Table 6:** Object/Color sets in the Putting experiment.

to solve any task we presented. For the objects we used simple, easily-distinguishable shapes, such as circles, squares, triangles, plus signs and x shapes etc.

**Finding & lifting:** For the comparisons to 3D (section 4.2), for the finding comparison, and the lifting trials in the putting task, the agent was rewarded if it moved onto the correct object (i.e. the one that matched the instruction), and the episode was terminated without reward if it moved onto the wrong object. For the putting trials, when the agent moved onto the object, it would pick the object up and place it on its head, so that the object was seen in place of the agent, analogously to how the background is obscured by an object carried in the 3D world. If the agent moved onto an object while carrying another, the object it was currently carrying would be dropped on the previous square, and replaced with the new object. If the agent moved onto the bed (striped lines in the screenshot) with an object, the episode terminated, and the agent was rewarded if the object was correct. See appendix D for some variations on this, including making the agent visible behind the object it is carrying, to ensure that this lack of information was not the factor underlying the results.

**Language lesion:** For the language vs. no-language comparison (section 4.5), if the agent moved onto a correct object, it received a reward of 1. This reward was both used as a training signal, and given as input on the next time-step, to help the agent figure out which objects to pick up. In addition, if the agent picked up an incorrect object for the current episode, the episode would terminate immediately, without any additional reward.

## C  AGENT DETAILS

### C.1  ARCHITECTURE PARAMETERS

The agent **visual processor** is a residual convolutional network with $64, 64, 32$ channels in the first, second and third layers respectively and 2 residual blocks in each layer.

| Sets | |
|---|---|
| $X_1$ (small) | basketball, book, bottle, candle, comb, cuboid block |
| $X_1$ (medium) | basketball, book, bottle, candle, comb, cuboid block, fork, football, glass, hairdryer, headphones, knife, mug, napkin, pen, pencil, picture frame, pillar block, plate, potted plant, roof block, scissors, soap, soap dispenser, sponge, spoon, table lamp, tape dispenser, toothbrush, toothpaste, boat, bus, car, carriage, helicopter, keyboard, plane, robot, rocket, train, vase |
| $X_1$ (large) | chest of drawers, table, chair, sofa, television receiver, lamp, desk box, vase, bed, floor lamp, cabinet, book, pencil, laptop, monitor, ashcan, coffee table, bench, picture, painting, armoire, rug, shelf, plant, switch, sconce, stool, bottle, loudspeaker, electric refrigerator, stand, toilet, buffet, dining table, hanging, holder, rack, poster, wall clock, cellular telephone, person, cup, desktop computer, mirror, stapler, toilet tissue, table lamp, entertainment center, flag, printer, armchair, cupboard, bag, bookshelf, pen, soda can, sword, curtain, fireplace, memory device, battery, bookcase, pizza, game, hammer, chaise longue, food, dishwasher, mattress, pencil sharpener, candle, glass, ipod, microwave, room, screen, lamppost, model, oven, bible, camera, cd player, file, globe, target, videodisk, wardrobe, calculator, chandelier, chessboard, mug, pillow, basket, booth, bowl, clock, coat hanger, cereal box, wine bottle, |
| $X_2$ | tennisball, toilet roll, teddy, rubber duck, cube block, cushion, grinder, racket |
| Color | aquamarine, blue, green, magenta, orange, purple, pink, red, white, yellow |

**Table 7:** Object/Color sets in the Negation experiment.

| Attribute | Sets | |
|---|---|---|
| | $\mathbf{s}$ | $\hat{\mathbf{s}}$ |
| Shape (S) | tv, ball, balloon, cake, can, cassette, chair, guitar | hat, ice lolly |
| | $\mathbf{c}$ | $\hat{\mathbf{c}}$ |
| Color (C) | black, magenta, blue, cyan, yellow, gray, pink, orange | red, green |

**Table 8:** Object sets in the 2D x 3D experiment. The same sets of objects were used for DeepMind Lab and our 2D environment.

In the **learning algorithm**, actors carry replicas of the latest learner network, interact the with environment independently and send trajectories (observations and agent actions) back to a central learner. The learning algorithm modifies the learner weights to optimize an actor-critic objective, with updates importance-weighted to correct for differences between the actor policy and the current state of the learner policy.

# D   SOME CONTROL EXPERIMENTS FOR THE 2D VS. 3D PUTTING TASK

We outline some control experiments we ran to more carefully examine what was causing the difference in generalization performance on the putting task between the 3D environment and the different conditions in the 2D gridworld.

## D.1   GRID-WORLD EGOCENTRIC SCROLLING EFFECT VS. PARTIAL OBSERVABILITY

In the main text, we noted that introducing egocentric scrolling to the grid-world putting task resulted in better generalization performance (although still worse than generalization performance in the 3D world). We attributed this effect to the benefits of an egocentric frame of reference. However, there is another change that is introduced if an egocentric frame of reference is introduced by the size of the visual input is kept the same – the world becomes partially observable. If the agent is in one corner of the room, and the bed is in the opposite corner, the bed will be outside the agent's view in the egocentric framework. This increases the challenge of learning this task, and indeed the agent trained with egocentric reference took longer to achieve good training performance, despite generalizing better. Thus, an alternate interpretation of the increase in generalization performance would be that this partial observability is actually contributing to robustness by forcing the agent to more robustly remember things it has seen.

In order to examine this, we ran two additional experiments, one with an egocentric scrolling window that was twice as large as the room (in order to keep everything in view at all times), and another with a fixed frame of reference of this same larger size. The latter was a control because, for memory reasons, we had to decrease the resolution of the visual input slightly with this larger view window ($7 \times 7$ pixels per grid square instead of $9 \times 9$). Test episode performance in the case of the fully-observable egocentric window ($M = 0.64$, $SD = 0.04$) was not significantly different from that in the ego-centric and partially-observable case ($M = 0.60$, $SD = 0.14$), $t(8) = 0.64$, $p > 0.05$. We therefore conclude that it is invariant frame of reference afforded by the egocentric perspective, rather than partial observability per se, that leads to better generalization in this case.

## D.2   SCROLLING AND AGENT VISIBILITY WHEN HOLDING OBJECTS IN THE GRID WORLD

In the experiments reported in the main text, when an agent picks up an object in the grid world, it places the object on its head, and so the object obscures the agent's location. This was meant to mimic the way that an object held in front of the agent in the 3D world blocks a substantial portion of its field of view. However, this could also make the task more difficult for the agent, because it is required to use its memory (i.e. its LSTM) to recall what its location is when it is carrying an object – it is no longer visible directly from a single observation, since the agent could be under any of the objects on screen. By contrast, in the egocentric scrolling perspective, the agent is always in the center of the screen, so this ambiguity is removed. This could provide yet another explanation for the difference between the fixed view and scrolling view.

To examine this, we ran an additional experiment, where we made the agent visible in the background behind the object it was carrying. That is, since the agent is a white square, effectively an object it is carrying now appears to be on a white background. However, test episode performance in this case ($M = 0.38, SD = 0.07$) was, if anything, worse than when the agent is completely hidden behind the object ($M = 0.42, SD = 0.14$). This is somewhat surprising at first glance, since in the 3D world the agent seems to be able to generalize to e.g. finding the bed when objects it has never carried to the bed are obscuring part of its view. However, it reflects the broad strokes of our argument – in the grid-world, seeing examples of carrying only three objects on top of the agent may not suffice to induce a general understanding of how to parse an agent in the background from the object on top of it. By contrast, in the 3D environment the agent will see many perspectives on each object it carries, and will see the background obscured in many different ways from many different perspectives over the course of its training episodes. We suggest that this leads to more robust representations.

