# OpenReview forum: "Environmental drivers of systematicity and generalization in a situated agent"
_ICLR.cc/2020/Conference — Accept (Poster)_

### Official Review · AnonReviewer3 · 2019-10-20
**Official Blind Review #3**

**Rating:** 6

**Review:**

This paper studies systematic generalization in a situated agent. The authors examine the degree to which various factors influence systematic generalization, including 2D vs. 3D environments, egocentric vision,  active perception, and language. The experiments reveal that the first three factors, but not language, promote systematic generalization. The experiments are well-done and worthwhile, and identifying the key factors that affect generalization is a strength of the paper.

I have two main criticisms. First, the model's abilities for systematic generalization are overstated. Second, critical details about the experiments are omitted that make them difficult to evaluate.

Let's start with the abilities of the model. The title of the paper is "Emergent systematic generalization in a situated agent," which of course implies that the agent has "systematic generalization." The authors go on to say, in the abstract, that "we demonstrate strong emergent systematic generalisation in a neural network agent". The results, however, fall short of these statements.

The strongest results pertain to generalizing a highly-practiced action such as "lifting" or "putting" to novel objects. In this case, highly-practiced means that the actions have been trained on 31 unique objects for millions of steps. However the paper does not study whether or not the agent can learn a novel action (e.g. "lifting" or "putting" with only a few examples) and generalize it systematically to familiar objects. Nor does it study whether novel actions can be combined systematically in new ways using relations and modifiers such as "finding the toothbrush ABOVE the hat" or "finding AND putting" or "putting to the right of." Benchmarks for systematic generalization such as SQOOP and SCAN include these types of generalizations, and an agent with systematic generalization should handle them as well. To be clear, I don't think it's necessary to add additional experiments to the paper, but the current results should not be overstated in their generality.

Even within the reported experiments, the results suggest that systematicity is lacking in several places. In the negation task, where chance is 50% accuracy, the agent achieves only 60% correct after learning from 40 unique words and 78% performance with 100 unique words (doesn't systematic generalization imply 100%?) For the putting tasks, the agent achieves 90% correct in one experiment (section 3.2) and then only achieves 63% correct in another (section 4.2). Again, the generalization abilities seem far from systematic.

Critical details about the action space and the simulation parameters are needed. The action-space has 26 actions, but the paper does not say what these actions are. These details are crucial to understanding what is required to generalize "lift" or "put" to new objects -- instead the paper only says that "in particular the process required to lift an object does not depend on the shape of that object (only its extent, to a degree)" and that "shape is somewhat more important for placement" compared to lifting.

I would consider updating my evaluation if the authors make revisions to ensure that the evidence supports their conclusions. The paper's title should also be supported by the findings; to offer a suggestion, something like "Richer environments promote systematic generalization in situated agents".

Other suggestions
- The axis on Fig. 2 is too small to read. Also, it should be mentioned in text that the network is trained for 100 million+ steps (also, what is a step? how many episodes was it trained for?)
- The number of objects in sets X_1 and X_2 is important and should be mentioned in the main text.

------

** Update to review **

Thanks for your response to my review. It's clear that the authors have made considerable effort to improve the paper. In particular, the revised title, abstract, and introduction now more accurately reflect the contributions of the paper. It's not perfect, but the paper is improved and I revised my score accordingly.

While it did not affect my final score, not all my suggestions were incorporated and I hope the authors will make further improvements in their revisions. The number of objects in sets X_1 and X_2  (Sections 3.1 and 3.2) are not mentioned and are tucked away in the appendix' this should be in the main text. Thanks for providing the list of 26 actions, but it's still not completely clear what makes a successful "lift" or "put" in terms of the sequence of actions. Finally, rather than simply saying that your agent "in no way exhibits complete systematicity" (Discussion), I hope the authors will expand on this and discuss the limitations of their experiments, and the kinds of systematicity not addressed and which could be the focus of future work.

**Experience Assessment:**

I have published in this field for several years.

**Review Assessment: Checking Correctness Of Derivations And Theory:**

N/A

**Review Assessment: Checking Correctness Of Experiments:**

I carefully checked the experiments.

**Review Assessment: Thoroughness In Paper Reading:**

I read the paper thoroughly.

---

> ### Author Response · Authors · 2019-11-12
> **Please verify that your concerns have been addressed.**
>
>
> Thank you for your review! We hope your main concerns are mitigated by the reframing described above. We have also thoroughly edited the remainder of the text thoroughly to make sure no other claims could be misconstrued in ways that you describe. We won’t list all minor edits here, but, as an example, in Section 1.1 we have added the sentence. Please also take a look at the revised manuscript.
>
> "Given that human reasoning is often not perfectly systematic (O.Reilly et al, 2013), here, we consider systematicity to be a question of degree, rather than an absolute characteristic of an intelligent system."
>
> And in Section 5, we have modified a sentence into the following:
>
> "We also emphasize that our agent in no way exhibits complete systematicity. The forms of generalization it exhibits do not encompass the full range of systematicity of thought/behaviour that one might expect of a mature adult human, and that none of our experiments reflect the human ability to learn (rather than extend existing knowledge) quickly (as in, e.g Lake et al, 2015)."
>
> We have also added a table with the action set to the appendix, and a short passage describing the implications of the action set (e.g. what behaviour is specifically required to lift and place an object).

---

> ### Author Response · Authors · 2019-11-15
> **Please let us know if you are happy with our resolution of your concerns**
>
> If you have further concerns in light of the revisions that we have made, we would really appreciate knowing this while there is still time to make improvements to the paper. Many thanks again for your efforts and engagement!

---

### Official Review · AnonReviewer2 · 2019-10-22
**Official Blind Review #2**

**Rating:** 6

**Review:**


=============================== Update after revisions =====================================================

In my initial review, I had raised some issues with the interpretation of the results and suggested some control experiments to tighten the conclusions. The authors chose to weaken their initial claims by rephrasing their conclusions instead. I understand that there may not have been enough time to run many of the experiments I suggested, but I still think they are worth considering for the future. I'm mostly satisfied with the rephrasing of the conclusions in the revised paper, so as promised, I'm happy to increase my score and recommend acceptance.

I spotted several typos in the revised paper, however: section 4.1: "we choose to consider negation ...", p. 5: "for for ...", a citation on p. 5 is not compiled correctly. There may be more. For the final version please make sure to go through the paper thoroughly a couple of times and fix all the typos.

========================================================================================================

The authors present a systematic study of generalization in agents embedded in a simulated 3d environment. I think there are some interesting results in this paper that might be useful for people to know about. I appreciate the thoroughness of the experiments, in particular. I have, however, some issues with the interpretation of several of the main results. I would be happy to increase my score if we can resolve some of these issues. Here are my main concerns:

1) In the experiments in section 3, only a limited test set is used. How is the train/test split decided in these experiments? Table 6 suggests that you have a much larger repository of objects. Why not use all possible objects in the test set? It is a bit premature to declare your results as systematic generalization if you can’t show that it actually works for a much larger set of test objects (ideally for all possible objects).

2) Section 4.1: in these experiments, the training set size is increased, but the test set size is kept constant (and small), so the train/test size ratio also increases. So, an alternative explanation of the results in this section is that the model behaves largely according to visual similarity and as the training set size is increased, it becomes easier to find a training set object that is visually similar to any test set object. I think the authors should run an experiment where both training and test set sizes increase by the same amount so that the train/test set size ratio stays constant. If the model can’t achieve systematic generalization in that case, it would be wrong to conclude, as the authors do now, that increasing the training set size itself improves systematic generalization. The correct conclusion would rather be that increasing the train/test size ratio improves generalization, which is a weaker conclusion. Please note that the results in this section are quite similar to those in Lake & Baroni (2018) and in Bahdanau et al. (2018) (see their Figure 3). Bahdanau et al. (2018), for example, also show that increasing train/test set size ratio (their “rhs/lhs” ratio) improves generalization in generic neural networks. It is interesting to note, however, that neither Lake & Baroni (2018) nor Bahdanau et al. (2018) interpret these results positively (i.e., these results don’t show systematicity), whereas the current paper seems to put a more positive spin on essentially the same result. I think these earlier results should be explicitly discussed here and the authors should justify why they are interpreting the results differently (if they are). It should also be noted that in the real world the train/test size ratio for humans is presumably very small, perhaps zero (given the compositional abilities of humans).

3) Section 4.3: I don’t think the results in this section are sufficient to establish the egocentric frame per se as the key factor. One possibility is that perhaps the frame doesn’t have to be centered on the agent, but as long as it has some systematic relationship to the agent’s location (for example, the center of the visibility window could be some distance away from the agent, and the agent itself may or may not be inside this window), that’s good enough to get generalization improvements. An even weaker possibility is that simply a moving frame is enough for improved generalization. In this case, the reference frame doesn’t even need to have a systematic relationship to the agent’s location. For example, the frame could be relative to a fictitious agent that randomly explores the environment. I think the authors should run some experiments to rule out these possibilities if they want to claim that the egocentric frame itself is responsible for generalization improvements.

4) Section 4.4: In the experiments in this section, I think there are two relevant factors that need to be better disentangled: 1) the number and variability of image frames experienced by the two models; 2) the active perception aspect (the fact that the agent interacts with the environment and affects its own perceptual experience in one case). The authors claim the second factor as the key aspect enabling better generalization, but 1) is equally likely (this would be more in line with a standard data augmentation type result). A good control experiment here would be to not just use the first frame but a larger number of more variable frames for training the non-situated agent (for example, one can use image frames that would be seen by a camera that more or less randomly moves in front of the objects perhaps with the constraint that both objects are always at least partially visible). If the classification model generalizes as well as the situated agent in this control condition, you cannot claim active perception as the key factor.

5) As a more general point, it’s a bit frustrating to have to judge systematic generalization by only looking at the results of some limited set of experiments. How do I interpret the results if the agent achieves only 84% accuracy in some experiment (as opposed to 100%)? It would be much better if the authors could somehow more rigorously prove systematicity. Here, I don’t necessarily mean “prove” in a mathematical sense, but just analyzing the learned representations a bit more rigorously and being able to say something along the lines of: here’s exactly how the trained model represents “lift”; because of reason X, Y, Z, this representation is completely disentangled from all object representations in the dataset (and ideally from all possible object representations, because that’s really what true systematicity entails, although I highly doubt that any generic model of the type studied in this paper will be able to achieve this, regardless of the amount and type of input it receives).

More minor issues:

6) In Table 5, “table lamp” appears both in training and test sets. Is this a typo?

7) Some results are presented in the appendix without any mention in the main text (Appendix D. 2). I think this is not a good practice in general. In the main text, please make sure to mention, however briefly, every result that appears in the appendix (something along the lines of "This result could not be explained by confound X or Y (Appendix Z)" would suffice).

8) Font size in Figure 2 is tiny (axis labels are impossible to read), please make it bigger. You don’t need that many ticks on the axes.

**Experience Assessment:**

I have read many papers in this area.

**Review Assessment: Checking Correctness Of Derivations And Theory:**

N/A

**Review Assessment: Checking Correctness Of Experiments:**

I carefully checked the experiments.

**Review Assessment: Thoroughness In Paper Reading:**

I read the paper at least twice and used my best judgement in assessing the paper.

---

> ### Author Response · Authors · 2019-11-12
> **Please verify that your concerns have been addressed**
>
> Thank you for your review! Please take a look at the revised manuscript to verify that your concerns have been addressed.
>
> 1) The more objects we use, the longer the agent takes to learn the training task so we didn’t work with all of them. Comparing the 4.2 and 3 we see that generalization on the ‘putting’ task is better with more objects in the training set (Fig 2b vs Table 2). We are sure that the effect would only be stronger if we ran the experiment with more objects than currently in 3 (poor performance if the number of objects involved during training is large is certainly not a failure mode of this agent!). The objects in the train/test set were chosen at random.
>
> 2) We agree that 4.1 shows that that *for a given size of test set* increasing the size of the training set improves generalization (or, equivalently, increasing the train-to-test ratio). We have edited two sentences in the paper to make this clear, and to link to the passage in Bahdanau et al. 2018. The experiments do not provide information about how generalization changes when the ratio stays the same but the size of both increases; do you have a link to literature / theory for why this is an interesting thing to investigate?
>
> 3) We agree that our experiments in 4.2 say nothing about whether the effects of what we call “ego-centric” perspective rely on the camera being centred on (rather than just tied to) the agent, nor if the effect might be the same if the camera was just moving randomly. Indeed, due to the spatial invariance afforded by the convolutional architecture, it's likely that the centering is less important than the fact that the agent is in some consistent location in the visual input. We cannot say if the effect might be the same if the camera were moving randomly, but this does not seem to have the same ecological validity or intuitive basis for investigation as an egocentric or allocentric frame of reference. We have added a sentence which should hopefully temper these claims: “This suggests that the visual invariances introduced when bounding the agents perspective, which in our case was implemented using an ego-centric perspective, may improve an agent's ability to factorise experience and behaviour into chunks that can be re-used effectively in novel situations.” We have also changed the title of the section to simply “Visual invariances in agents' perspectives” rather than “Egocentric frame of reference”, and the appropriate sections in the abstract (e.g.,”...the visual invariances afforded by the agent's perspective, or frame of reference”)
>
> 4) We agree that the experiment does not allow distinction between interaction (RL) and merely learning from a video. To make this clearer, we have changed the term “Active perception over time” to “Temporal aspect of perception”. Our 3D environment does not have the functionality to enable the control experiment that you describe (e.g. guaranteeing objects in view), and it would be very hard to control for the various factors at play (the length and quality of the sequences of frames and interaction with the world are essentially entangled in some way). We would like to explore this question further in future projects, but for now we will make the conclusions tighter.
>
> 5) We follow Fodor and Pylyshin’s definition of systematicity in terms of what ‘thoughts’ can be ‘understood’. As we understand it (see discussion above), systematicity is not defined in terms of internal representations (some work has argued that disentangled representations should lead to better systematicity, but this has not been conclusively shown empirically as far as we know). We make no claims about the internal representations of our agent in this work. In any case, thorough analysis and interpretation of internal representations in models is very hard (an active research area) and beyond the scope of this experimental study.
>
> 6) Yes, well spotted - it should only be in the test set. Amended
>
> 7) Noted and amended
>
> 8) Noted and amended

---

> ### Author Response · Authors · 2019-11-15
> **Please let us know if your concerns have been addressed**
>
> If you have further concerns in light of the revisions that we have made, we would really appreciate knowing this while there is still time to make improvements to the paper. Many thanks again for your efforts and engagement!

---

> > ### Comment · AnonReviewer2 · 2019-11-15
> > **"fine-grained" control**
> >
> > I appreciate the revisions. I have yet to read the revised paper more carefully, but upon a quick read, I noticed that in several places you claim that your tasks require "fine-grained" control. I disagree with this claim. The control required in your tasks is extremely coarse (compared to controlling a robotic hand, for instance) and this in fact seems to be crucial for generalization to work in your tasks: for instance, what the agent needs to do to lift an object isn't very sensitive to the precise shape of the object. This point is explicitly acknowledged on p. 4 (last paragraph of section 3), and I appreciate that. But then references to your tasks requiring "fine-grained control" elsewhere in the paper become misleading. So, I would suggest that you remove those references from the paper (doing a quick ctrl+F, I could find two such instances on p. 2 and p. 3 respectively).

---

> > > ### Author Response · Authors · 2019-11-15
> > > **Noted, and thank you**
> > >
> > > Thank you again for continued engagement with our paper. We agree that fine-grained isn't the  correct term and have removed this as you suggest.

---

### Official Review · AnonReviewer4 · 2019-11-05
**Official Blind Review #4**

**Rating:** 6

**Review:**

This work studies factors which promote combinatorial generalization in a "neural network agent" embodied in a 3d simulation environment. The authors present interesting experiments and some insightful empirical findings on how a richer environment and a first-person egocentric perspective can aid a simple neural net to generalize better over previously unseen tasks. While I truly commend the effort undertaken to perform the experiments, I have several concerns which I explain below and would be happy to raise my score if they can all be addressed satisfactorily:

1) While the authors interpret the experiment results in sec 4.1 in a positive way, the results don't seem to necessarily indicate good systematic generalization. For instance, after learning with 40 words the agent only achieves 60% test accuracy. While the accuracy increases to 78% on training with 100 words, the training and test accuracy gap indicates that the performance is still far from any kind of systematic generalization. The results instead seems to be hinting that neural nets don't indeed perform combinatorial generalization on their own, but can be forced towards it by supplying them huge amounts of diverse data (which is not true for humans). Also, the fact that increasing the number of words helps in generalizing better is true for most ML models and does not come as a surprise. So the results in this subsection are somewhat trivial and do not necessarily contribute any new understanding.

2) For the experiments regarding egocentric frame in sec 4.3, I feel that the results are not really conclusive (even including the control exps in appendix D). Could it be that if one uses any frame rigidly attached (i.e. fixed displacement and rotational coordinates) to the agent's egocentric frame, one would achieve the same generalization performance? It is also possible that as suggested by authors in sec 4.4, it is just the motion of the egocentric frame which might be giving diverse views of the environment to the agent. So the frame might not even need to be egocentric, but just a moving frame which gives richer and diverse views whenever the agent moves. Please include experiments to test for these possibilities.

3) In section 4.4, the authors have trained the non-embodied classifier with just a single image frame. But this does not necessarily justify the conclusion that active perception helps in generalization. This is because the motion of the RL agent gives it both a varied set of views AND also control over what views to obtain by taking actions. In order to better understand which of these factors (or perhaps both) aid in generalization, another set of experiments is required which shows the classifier agent more images while keeping the desired object in view. In one experiment, these images should be chosen with random movements but the number of such images provided to the classifier should be increased in sub-experiments to gauge if giving more varied views bridges the performance gap between the classifier and the RL agent's generalization performance. In a second experiment, one might want to first train the RL agent, then extract a few (say 10) frames out of its enacted policy for all pairs of objects and use these frames as a part of the training set for the classifier agent. This would allow one to gauge if both varied views and actively selecting to interact with the environment can help bridge the generalization gap.

4) Lastly, sec 4.5 seems to be hinting at a potentially very incorrect conclusion: "language is not a large factor (and certainly not a necessary cause) of the systematic generalisation...". This cannot be said from the small single experiment presented in sec 4.5. For instance, that experiment has been devised in a way that an optimal policy can be found with/without language. However, if a language input is provided to explicitly state the desired object, that might speed up the training of the RL agents significantly. In such a case, it might be helpful to see if learning the policy with the language input is being accomplished with a much lower number of frames during training, as opposed to when no language input is provided. Please provide the training error plots. But regardless of the plots, the experiments can still be quite inconclusive since language helps in systematic generalization in a variety of other ways apart from what has been tested for. In general, language starts helping humans once it has been acquired to a sufficient extent since one needs noun-concept linkages, verb-action linkages etc. to have been acquired a priori before the benefits of language emerge in combinatorial generalization. Training an LSTM to understand the language commands in tandem with learning policies for picking desired objects could lead to sub-optimal or heavily over-fitted language models which may not help in generalization. Testing for the true role of language will require many more experiments, which may be somewhat out of scope for this paper given the space constraints for a single paper. But, I would advise the authors to refrain from drawing hasty inferences about the role of language without thorough experimentation.

Minor issues:

1) What are the 26 actions in the Unity 3D environment in section 3? It is important to know the action space to understand how easy or hard it is for the agent to learn generalizable policies.
2) The x-axis of Figure 2 is not readable at all. Please rectify those graphs and reduce the number of ticks.


-------------------------- Update after interaction during author feedback period -------------------------------
I appreciate the efforts that the authors have undertaken to address my concerns. While the paper is far from perfect, it is still a very thought provoking work and I believe that it would make a valuable contribution to the line of works on systematic generalization in embodied agents. I am updating my score to reflect the same.

**Experience Assessment:**

I have read many papers in this area.

**Review Assessment: Checking Correctness Of Derivations And Theory:**

N/A

**Review Assessment: Checking Correctness Of Experiments:**

I carefully checked the experiments.

**Review Assessment: Thoroughness In Paper Reading:**

I read the paper at least twice and used my best judgement in assessing the paper.

---

> ### Author Response · Authors · 2019-11-12
> **Please verify that your concerns have been addressed in the revised manuscript.**
>
> Thank you for your review! Please take a look at the revised manuscript to verify your concerns have been addressed.
>
> 1) You are right that the main finding of 4.1 is should not be surprising (this is why we wrote “the fact that larger training sets yield better generalization in neural networks is not novel or unexpected” in the section). However, we find that the context in which it is shown (negation, a problem with a long history in neural net research, and an operator which is, in some sense, maximally non-compositional) is interesting, to us at least.
>
> 2) We agree that the experiments in 4.2 say nothing about whether the effects of what we call “ego-centric” perspective rely on the camera being centred on (rather than just tied to) the agent, nor if the effect might be the same if the camera was just moving randomly. All we claim is that ‘if the window is centred on the agent (or the agent has first person perspective in 3D) then generalisation improves’. We will add a sentence to make this clearer.
>
> 3) You are correct that the experiment does not allow distinction between interaction (RL) and merely learning from a video (see also our response to Reviewer 2). To make this clearer, we have changed the term “Active perception over time” to “Temporal aspect of perception”. We would like to explore this question further in future projects / when possible.
>
> 4) We agree that the conclusion that you cite as "language is not a large factor (and certainly not a necessary cause) of the systematic generalisation..." would be entirely unwarranted based on this experiment. However, the complete sentence from which those words are taken reads "While not conclusive, this analysis suggests that language is not a large factor (and certainly not a necessary cause) of the systematic generalisation that we have observed emerging in other experiments." We don’t think this is a hasty or unwarranted conclusion given the evidence, but would be happy to discuss further. To remove any room for doubt about this, we have changed it to "While not conclusive, this analysis raises the possibility that language may not be playing a significant role (and is certainly not the unique cause) of the systematic generalisation that we have observed emerging in other experiments"
>
> We have also added a description of the action set to the appendix and fixed up Figure 2.

---

> > ### Comment · AnonReviewer4 · 2019-11-13
> > **More experiments/results needed to address my concerns**
> >
> > I appreciate your response and the clarifications provided but I want to emphasize that in an experiment-focused work like this one, it is important to go the extra mile and disentangle the effects of the factors being studied to as much an extent as possible. Currently, while the claims you make are technically correct and are now also worded appropriately, the factors that are being studied are in some sense "compound factors" and have not been disentangled appropriately.
> >
> > For instance, for my concern 2), a frame always centered on an agent in the 2D grid world is a frame which:
> > (a) masks some information far away from the agent,
> > (b) is fixed relative to the agent's coordinates, and
> > (c) moves as the agent moves.
> > By looking at the results in the paper, any reader would agree that such a frame improves generalization, but it is still unclear in what proportion do the above three features contribute in improving the generalization performance. I acknowledge your efforts in performing control experiments to study the masked information part (partial observability), but the effects of motion still remain to be disentangled from others via appropriate control experiments.
> >
> > Similarly for concern 3), just choosing to change the section names to be technically correct, leaves the contribution of random motion vs intentional motion unclear and thereby leaves significant room for improvement in the paper.
> >
> > For concern 1), I understand that negation is a harder operator to generalize over since it is significantly non-compositional, but since we agree that using more training data improves generalization, is it really that surprising/interesting if the generalization on the negation operator improves somewhat? The test accuracy doesn't seem to be increasing all the way to the training accuracy which re-affirms the fact that negation can be hard to generalize on, but how does one assess that the amount of generalization observed by including more words in training was more interesting than one would have expected for other operators? Lacking this assessment, I'm still somewhat unsure about the utility of Sec 4.1. At least a clarification about why this section is interesting is needed to understand the contribution of this section.
> >
> > For concern 4), I am not misunderstanding your sentence or trying to interpret it out of context. My concern is that language benefits generalization in ways that cannot really be explored in the context of current work (more details in my first post on this). While I understand the limited context in which you are experimenting for the role of language here, I also understand that the task is solvable with/without language. In other words, it is not surprising for an RL agent to learn an optimal policy even without language, given enough frames/trajectories for training (disregarding trajectories where the agent chooses the first object incorrectly). However, is it possible to show how much training experience was required (in terms of frames and/or trajectories) before the language-based and vision-only agents achieved this level of generalization? Did the use of language commands increase or reduce the amount of experience required to generalize to the extent shown? Providing this information would be very useful since it would let the reader know if language helped speed-up (or slow-down) convergence to the final generalization performance in terms of the amount of experience required, while not necessarily effecting the performance level at convergence.
> >
> > Overall, I really like the effort that the authors have undertaken to perform the initial experiments and to re-write certain sections of the paper. But there is currently significant scope of improvement in the paper in terms of the experiments performed. I would be happy to accept the paper once the above concerns have been addressed.

---

> > > ### Author Response · Authors · 2019-11-14
> > > **Reason why the proposed new experiments would not resolve important questions**
> > >
> > > We understand your views. To be clear, we are not trying to avoid running new experiments. However, we simply cannot see a way to practically disentangle the factors further, and also are a little unsure about the ecological or theoretical value of doing so (even if it were possible). To illustrate the issue we're facing, consider factors (a-c) in your reply (thank you for laying these out!). As you acknowledge, we have ruled (a) out as contributing to the effects that we observe in the appendix. What exactly is the experiment that would separate (b) and (c)? Clearly, since if the agent moves, its coordinates will move, (b) implies (c). So we need a situation in which (c) but not (b). This would require an environment in which a field of view was centred on the agent, and moved when the agent moved, but moved in a direction that was random (or not quite random?) compared with that of the agent. But what debate, scientific question or model of animal learning would this result inform us about beyond what we have already showed? For any cause and any effect, one might go in search of finer-grained causes, but it feels to us like this should only be done if there is some theory or wider reason for finding a particular level of analysis or explanation important.
> > >
> > > Taking a step back, relative to the other contemporary literature in this space, our results are the first empirical results that demonstrate the effects of more naturalistic environmental factors (coarse they may be) on an agent’s systematic generalization. Our results are a direct consequence of recent work exploring these questions in completely abstract domains, where even the “coarse” factors studied here were not explored at all, or were impossible to explore by design.
> > >
> > > For concern (3), we are a bit more clear on the precise question you would like us to explore. We share your view that it would be nice to somehow disentangle the effect of intentional motion and interaction over time with passively modelling a scene with a temporal aspect. However, when designing the details of the experiment to run, things very quickly get murky. How exactly should we record a passive view of two objects that was in some sense neutral with respect to the agent's policy? The only option that we can think of would be a camera that orbits the objects at a fixed distance and moves its lens to focus on the objects in question. Even if it were possible for us to set this up, it feels quite a contrived situation that has limited ecological or practical validity. Indeed, it would not definitely answer the question - one could continue to doubt the outcome by questioning the radius of the orbit or the control program that we must implement for moving the lens to fixate on the objects, or other such design decisions.  Given these issues, we feel that by far the most important modification that we have made to the paper is that the claims and conclusions that we reach are now entirely aligned with the facts of the original experimental effects that we observe.

---

> > > > ### Comment · AnonReviewer4 · 2019-11-14
> > > > **Some concerns resolved**
> > > >
> > > > Alright, I consider that a good explanation :)
> > > >
> > > > I have also personally given more thought as to how to separate the finer-grained effects for my concerns (2) and (3), but also couldn't devise good practical experiments to do so (without either coming up with heavily contrived situations or introducing even more new factors which would then need to be disentangled further). So at this point I will consider these concerns resolved.
> > > >
> > > > I still request the authors to please provide a good justification for how exactly to assess sec 4.1 results as being interesting enough (concern 1) and provide the required plots for concern (4).

---

> > > > > ### Author Response · Authors · 2019-11-15
> > > > > **Experiments re-run and passages edited to resolve final concerns**
> > > > >
> > > > > Thank you. Regarding your concern (1), the topic of whether neural networks (or connectionist models) can learn an adequate treatments of logical operators has a long history, which can be traced back to the PDP books (as noted in the paper), discussed at length by Steedman [1]. These treatments consider the ability learn to approximate negation symbolic inputs like logical expressions or natural language. Here we extend these analyses by considering a fully-situated, behavioural metric for the comprehension of negation, and consider ability to generalize as evidence of an adequate representation. To further underline why this may be of interest to the research community, consider the following proposal from Steedman (who is certainly not an avowed connectionist) for a situated model of language processing.
> > > > >
> > > > > =================
> > > > >
> > > > > It is likely that such a research program would proceed by first conceptualizing primarybodily actions and sensations, then coordinating perception and primary actions likereaching, then conceptualizing identity, permanence and location of objects, first independent of their percepts, then of the particular actions they are involved in, amounting tothe internalization of the components of a stable world independent of the child’s actions.Later stages would have to include the conceptualization of more complex eventsincluding intrinsic actions of objects themselves (such as falling), translations and eventsinvolving multiple participants, intermediate participants including tools, and goals. Atthis final stage of purely sensory-motor development most of the prerequisites forlanguage learning would be established, perhaps embedded in RAAM or some otherassociative memory, and could be used to support a program of inducing a similarlylayered sequence of linguistic categories such as: deictic terms based on a proximal/distaldimension (whose central place in language development with respect to reference anddefiniteness is discussed by Lyons, 1977—cf. Freud, 1920, pp. 11–16 for a revealing casestudy), markers of topic, comment and contrast, common nouns, spatial and path terms,causal verbs, modal and propositional attitude verbs, and temporal terms. It is likely thatthe semantic theory that would emerge from this work would be rather unlike anythingproposed so far within standard logicist frameworks. Such a semantics would be likely tomake us view phenomena like quantification, modality, negation, and variable-binding innew ways, within a unified theory combining symbolic and neurally-grounded levels" [1, p630]
> > > > >
> > > > > ==============
> > > > >
> > > > > More practically, we started on negation because our agent was generalizing very well on other tests of systematicity, and we wanted to consider the limits of the experiential approach we advocate here. We hope that including an experiment where our best agent is imperfect may stimulate new ways to improve systematicity, either through environmental or agent-based methods; indeed, when we have shared this work with others, many have been most engaged with trying to improve on this aspect.
> > > > >
> > > > > In order to express these sentiments more clearly we have added the following sentences:
> > > > >
> > > > > "Of course, the mere fact that larger training sets yield better generalisation in neural networks is not novel or unexpected. On the other hand, we find the emergence of a logical operator like negation in the agent in a reasonably systematic way (noting that adult humans are far from perfectly systematic (Lake et al. 2019)), given experience of 100 objects (again, not orders of magnitude different from typical human experience), to be notable, particularly given the history of research into learning logical operators in connectionist models and the importance of negation in language processing [Steedman, 1999]."
> > > > >
> > > > > "We choose to consider negation it is an example of an operator on which we found that, for our standard environment configuration, our agent unequivocally fails to exhibit an ability to generalize in a systematic way."
> > > > >
> > > > > Regarding your concern (4), we have re-run the experiment to save the learning curves, and put a plot of these dynamics into the final figure in the paper. As you suggest, generalization does start to take-off more quickly in the language condition. Interestingly, a large amount of training the two conditions converge on the test trials, even though a small gap remains on the training episodes. We have updated the conclusions in that section too.
> > > > >
> > > > > We hope that our efforts resolve your final concerns. Thank you for your engagement with the paper; it has helped to improve it substantially. We hope you agree that it's now in shape to make a valuable contribution to the growing literature and debate on generalization and representation in embodied agents.

---

> > > > > > ### Comment · AnonReviewer4 · 2019-11-15
> > > > > > **Concerns resolved and change of score**
> > > > > >
> > > > > > I appreciate the efforts that the authors have undertaken to address my concerns. While the paper is far from perfect, it is still a very thought provoking work and I believe that it would make a valuable contribution to the line of works on systematic generalization in embodied agents. I will be updating my score to reflect the same shortly.
> > > > > >
> > > > > > Minor feedback:
> > > > > > In the newly added plot in figure 5, there are four curves annotated in the legend (light green, light black, dark green and dark black). But apart from those, I can also see several other very lightly colored curves in the actual plot. Can you clarify what those are (or remove them if they are not needed)?

---

> > > > > > > ### Author Response · Authors · 2019-11-15
> > > > > > > **New curves were traces of replicas in each condition**
> > > > > > >
> > > > > > > Thanks for revising this so quickly! The light curves were an artifact of the plotting library; each dark line was a mean over multiple agent replicas, which were still rendered very lightly. We have removed these and uploaded a new version.

---

### Author Response · Authors · 2019-11-07
**Clarification on the central point of divergence and proposed new title/abstract**

Thank you for your thoughtful reviews. Based on these comments, we believe we can move towards a version of the manuscript that would be more acceptable for you. Before responding to the finer points, we hoped to discuss what we think is the single clearest perceived limitation of our work. This involves the idea that our agent exhibits ‘complete’ systematicity (forgive loose terminology here) -- i.e. we account for all of the ways in which a human might generalise in a systematic way. To be clear, this is certainly not a claim we intended to make. In particular, we consider systematicity to be a question of degree rather than absolute, and the strongest claim we intend is that in several very specific circumstances we observe an agent exhibiting *more* systematic behaviour than in other circumstances. That is, we have revealed a relative difference in systematicity across training conditions. We tried to be explicit about this in the final paragraph with the sentence:

“We also emphasize that our results in no way encompass the full range of systematicity of thought/behaviour that one might expect of a mature adult human, and that none of our experiments reflect the human ability to learn quickly”

However, we have the impression that the title, abstract and intro are the main contributors to this misunderstanding. If you agree, we therefore propose changing the title and abstract as follows (and will edit the introduction along the same lines).

===================

Title: Environmental drivers of systematicity in a situated agent

The question of whether deep neural networks are good at generalising beyond their immediate training experience is of critical importance for learning-based approaches to AI. Here, we consider tests of systematicity that require an agent to respond to never-seen-before instructions by manipulating and positioning objects in a 3D Unity simulated room. We first describe a comparatively generic agent architecture that exhibits strong performance on these tests. We then identify three aspects of the training regime and environment that make a significant difference to its performance: (a) the number of object/word experiences in the training set; (b) the invariances afforded by a first-person or agent-centric perspective; and (c) the variety of visual input inherent in the perceptual aspect of the agent’s perception. Our findings indicate that the degree of systematicity that emerges in neural networks can depend critically on particulars of the environment in which a given task is instantiated. They further suggest that the propensity for neural networks to behave in systematic ways can increase if, like human children, those networks have access to many frames of richly varying, multi-modal observations as they learn.


======================

Please let us know whether, with changes along these lines would resolve your concerns around this issue? We intend to answer and amend the finer points of the reviews as well, but felt it important to try to reach consensus on this central issue first.

---

> ### Comment · AnonReviewer2 · 2019-11-07
> **why not remove "systematicity" altogether?**
>
> Thanks for the clarification. I think the problem is that a lot of people (most people?) understand systematicity to be a binary, all-or-nothing property (myself included). This is certainly how Fodor & Pylyshyn (1988) originally defined the concept: "... the ability to produce/understand some sentences is intrinsically connected to the ability to produce/understand certain others." It is clear from the context what they have in mind is some kind of logical implication: if you understand "lift X" and "find Y", you cannot fail to understand "lift Y" (this is also why I suggested more rigorously "proving" systematicity in my review (point 5) instead of trying to infer it from a limited set of experiments). So I would personally prefer that you didn't use the word "systematicity" at all. To me, what you demonstrate in the paper is more accurately described as simply improved generalization or improved out-of-sample (or out-of-distribution) generalization.

---

> > ### Author Response · Authors · 2019-11-07
> > **Removing 'systematicity' from the title and abstract (and some wider discussion on systematicity)**
> >
> > Thank you for your rapid engagement. We have thought about it and, while we believe the work to address and contribute to the wider systematicity debate, we are happy to adopt your recommendation to remove mention of systematicity (excepting the final sentence, where the connection to our results is very indirect, below) in the title and abstract. We can ensure similar amendments in the paper itself:
> >
> > ===================
> >
> > Title: Environmental drivers of generalization in a situated agent
> >
> > The question of whether deep neural networks are good at generalising beyond their immediate training experience is of critical importance for learning-based approaches to AI. Here, we consider tests of out-of-sample generalization that require an agent to respond to never-seen-before instructions by manipulating and positioning objects in a 3D Unity simulated room. We first describe a comparatively generic agent architecture that exhibits strong performance on these tests. We then identify three aspects of the training regime and environment that make a significant difference to its performance: (a) the number of object/word experiences in the training set; (b) the invariances afforded by a first-person or agent-centric perspective; and (c) the variety of visual input inherent in the perceptual aspect of the agent’s perception. Our findings indicate that the degree of generalization that networks exhibit can depend critically on particulars of the environment in which a given task is instantiated. They further suggest that the propensity for neural networks to generalize in systematic ways may increase if, like human children, those networks have access to many frames of richly varying, multi-modal observations as they learn.
> >
> > ======================
> >
> >
> > For what it's worth (and at risk of descending into the weeds with an old and sticky debate) it's clear from the original papers that Fodor and Pylyshyn were making an empirical observation (i.e.., the fact that we can understand Mary loves John if we understand John loves Mary is an observation about humans that led to the hypothesis that our models should be systematic in a similar way). F&P do not set out a rigorous definition beyond this intuition that we treat “semantically similar contents” in “similar ways”. One can quickly see that their empirical observation is not without controversy, and indeed, it may ultimately not be representative of a universal phenomenon of cognition. For example, there are numerous known counter-examples, e.g., from the Stanford Encyclopedia of Philosophy: do those who understand “‘within an hour’ and ‘without a watch’ also understand ‘within a watch’ and ‘without an hour’”? Moreover, it is not entirely clear what “semantically similar” contents are, whether these must be learned, or are to somehow be known a priori (who determines them, if so?).
> >
> > So, to treat systematicity as an abstract binary property that can be “attained” by a model may not be a view that appropriately considers the controversy behind it. We instead take the view that there may be a lot of important factors that contribute to whether *we observe* a system to be systematic. Enumerating these factors, and exploring the ways in which models appear to be systematic when these factors exist is an important empirical research topic.  Examples of such factors may be previous learning, the surrounding context, implicit knowledge, and so on.
> >
> > We also note a very relevant quote from the Stanford Encyclopedia entry on this topic:
> >
> > “"Jansen and Watter note however, that the sensory-motor features of what a word represents are apparent to a child who has just acquired a new word, and so that information is not off-limits in a model of language learning. They make the interesting observation that a solution to the systematicity problem may require including sources of environmental information that have so far been ignored in theories of language learning. This work complicates the systematicity debate, since it opens a new worry about what information resources are legitimate in responding to the challenge. However, this reminds us that architecture alone (whether classical or connectionist) is not going to solve the systematicity problem in any case, so the interesting questions concern what sources of supplemental information are needed to make the learning of grammar possible."

---

> > > ### Comment · AnonReviewer4 · 2019-11-07
> > > **Systematicity may not be a binary property**
> > >
> > > While I am not an expert at psychological texts on systematicity, a point that I wanted to discuss was if systematicity is indeed believed to be a binary property. As I understand it, systematic generalization is not understood to be a purely binary concept and can develop/improve over time. This is also expressed in prior work in psychology [Vygotsky, 1987] which discusses young kids' inability to generalize and often hold "spontaneous concepts" (which can be self-contradictory when applied to different situations). As kids grow over time and are subjected to rich and diverse inputs from their environment, systematic generalization slowly emerges and improves over the childhood years. So it might be just fine to retain the word systematicity, unless it is being a cause of confusion to a majority of readers.
> > >
> > > [Vygotsky, 1987] Vygotski, Lev Semenovitch, Robert W. Rieber, Aaron S.. Carton, and Lev Semenovitch Vygotski. The Collected Works of LS Vygotsky: Problems of General Psychology. Plenum Press, 1987.

---

### Author Response · Authors · 2019-11-13
**Please note; paper still subject to improvement**

We just wanted to make clear, we have not quite finished revising the paper yet. By the end of the rebuttal period we will have comprehensively re-written the introduction to remove the focus on systematicity, in line with the new abstract and title.

Please also note that the title in Open Review (above) is no longer the title of the paper, but we cannot change that here.

---

> ### Author Response · Authors · 2019-11-14
> **Paper now fully amended for consistency with new title and abstract**
>
> Please see new manuscript

---

### Decision · Program_Chairs · 2019-12-19

**Decision:**

Accept (Poster)

**Comment:**

The paper studies out-of-sample generalisation that require an agent to respond to never-seen-before instructions by manipulating and positioning objects in a 3D Unity simulated room, and analyzes factors which promote combinatorial generalization in such environment.

The paper is a very thought provoking work, and would make a valuable contribution to the line of works on systematic generalization in embodied agents. The draft has been improved significantly after the rebuttal. After the discussion, we agree that it is worthwhile presenting at ICLR.